# The Impact of a Sustainable Economic Development Focus on the Real Exchange Rate in Saudi Arabia

## Sami Alabdulwahab

Department of Economics, College of Business Administration, King Faisal University, Al-Hofuf 31982, Al-Ahsa, Saudi Arabia; salabdulwahab@kfu.edu.sa

**Abstract:** Saudi Arabia has launched its vision for the next decade, which is branded Vision 2030. Sustainable economic development is the core of the Saudi Vision 2030, which has seventeen sustainability programs covering a wide range of Saudi economic sectors. The aim of this study was to examine the impact of the sustainable economic development focus on the real exchange rate in Saudi Arabia. The ARDL cointegration method was used to assess the existence of the long-run relationships among the variables. The Sustainable Development Index (SDI) was used as a measure of sustainable economic development. Sustainable economic development was found to have a negative impact on the real exchange rate in terms of both the long- and short-run dynamics. Moreover, the GDP was found to have a positive impact on the real exchange rate in the long run and to have the highest coefficient of the model. However, the money supply was found to have no impact on the real exchange rate, while oil rent was found to have a negative impact on the real exchange rate in terms of both the long-run and the first-moment short-run dynamics. Government expenditure was found to be insignificant in the long run and to have a positive significant impact on the real exchange rate in the short-run dynamics. Furthermore, the sustainability impact on the real exchange rate in Saudi Arabia has not been discussed due to the economic structure that relies on oil and the change toward a sustainable economy that occurred in the recent decade. The results provide insight into the potential future challenges faced by the Saudi economy as sustainability programs progress in Saudi Arabia.

**Keywords:** real exchange rate; sustainable economic development; ARDL cointegration; Saudi Arabia

## 1. Introduction

Saudi Arabia launched its Vision 2030 economic program on 25 April 2016, and sustainable economic development is the program's main objective. Like other Gulf Cooperation Council (GCC) countries, Saudi Arabia relies on oil revenue as a source of income. However, Saudi Arabia attended its first G20 meeting in 2008 at the Washington summit. Moreover, Saudi Arabia hosted the G20 meeting in 2020, and one of the presidency meetings focused on the net zero emission transition. The G20 members motivated the adaptation of the programs that vindicated the impact of this transition. However, Saudi Arabia has been invited to be a member of BRICS in their 2023 summit, where sustainability is playing a role in its agenda with the importance of having sustainability to be socially and environmentally responsible. On the other hand, Saudi Arabia has been ranked 94 out of 166 in the SDG index with 67.70% goal achievement. The Saudi Vision 2030 has been promoted as a structural change in the Saudi Arabian economy, and this structural change should affect different sectors of the economy in different directions. It is important to address the wide range of impacts of this change to have greater visibility of the challenges that Saudi Arabia's economy could face in the future. However, Vision 2030 has seventeen goals covering a wide range of concepts directly relevant to the United Nations Sustainable Development Goals. Those goals will have direct or indirect impacts on Saudi Arabia's economic activities, and such impacts can be assessed with the Human Development Index

(HDI) or the Sustainable Development Index (SDI). However, the SDI has an advantage over the HDI because it builds on the HDI formula, implements a threshold for the income per capita of a country, and divides its outcomes by two ecological indicators with ecological impacts, $CO_2$ emissions and material footprint. Therefore, the SDI can be used to conduct strong measurements of ecological efficiency. A country's economy comprises different simultaneously active elements that affect its current and future directions. One of these elements is the real exchange rate; like the rest of the world, Saudi Arabia pays close attention to its real exchange rate to ensure economic stability. However, government policies in the service of sustainable economic development may impact the real exchange rate in different ways, and understanding the impact of sustainable economic development on the real exchange rate can help a government mitigate the challenges associated with launching economic sustainability programs. The aim of the study is to examine the existence of the long-run relationship between sustainable economic development and real exchange rates in Saudi Arabia. However, the SDI will be used to assess the impact of sustainable economic development on the real exchange rate in Saudi Arabia, which is the spirit of the Saudi Vision 2030. There are two methods to assess the relationship among economics variables. These methods are Johansen cointegration and Autoregressive Distributed Lag (ARDL) cointegration. The choice between the two methods depends on the property of the data that satisfies one of these two methods' inquiries. Sustainability is a new arena in economics research, and the focus has been driven toward developed countries. This study will try to touch on the impact of sustainability on an important factor, which is the real exchange rate in the economy that counts as the biggest oil producer in the world. However, this study, unlike other studies, focuses on Saudi Arabia and assesses the relationship between local and international macroeconomics factors. Using SDI as a factor within the econometrics model will increase the credibility of the result since SDI is an external assessment that has been conducted by an external organization. Moreover, this study carries out a new side of research that has not been investigated within the recent Saudi Arabia economics research literature. Furthermore, Saudi Arabia's GDP is ranked 18th in the world based on the World Bank database. Saudi Arabia's responsibility toward high-world origination and the obligations toward its 2030 vision within the sustainability context, as well as its position as a major oil producer in the world, motivates this study to be the country's focus.

## 2. Literature Review

The exchange rate is affected by different theoretical and practical elements. The Quantity Theory of Money (QTM) characterizes the local demand for money, which affects the real exchange rate through three elements, including income. However, additional elements can affect the exchange rate according to econometric models. Ref. [1] developed and assessed an econometric model of the real exchange rate for oil-producing countries in the Middle East and North Africa (MENA). They used an ARDL panel regression framework to conduct estimations for the set of studied countries. They found that the money supply, GDP, government expenditure, oil prices, and the USA's externally financed debt per GDP had long-run relationships with the real exchange rate. However, income was found to have a negative impact on the real exchange rate in the short and long run. Ref. [2] used an estimated dynamic panel regression framework to examine the relationships between the real exchange rate and macroeconomic fundamentals in Latin America; the estimation was divided into three subperiods to compare the outcomes of the three estimations, and the periods were found to overlap with each other after omitting two decades from 1980 to 2000 (including one decade at the beginning of the period). The first estimation covered the period between 1980 and 2019, and the results showed that income growth, inflation, fiscal policy, and monetary policy had positive impacts on the real exchange rate. Furthermore, the periods covering 2000–2019 and 2010–2019 showed similar outcomes in which the impact of income on the real exchange rate was negative. On the other hand, [3] analyzed the impact of macroeconomic factors on the conversion of

the USD to the Chinese Yuan Renminbi. They applied the ARDL cointegration approach to annual time series data to examine the existence of long-run relationships among the studied variables. The results suggested the existence of long-run relationships, as the variables were found to be cointegrated with each other. Furthermore, the impact of the GDP on the real exchange rate was found to be positive, whereas the impacts of the other variables were negative. Ref. [4] examined the relationship between economic growth and the real exchange rate in India, and they found that there was a long-run relationship between the real exchange rate and economic growth in India. However, the relationship between the real exchange rate and economic growth was shown to be unidirectional. The relationships among the variables can also be tested with different variable transformation approaches. Ref. [5] conducted the expectation transformation of variables by analyzing the real-time survey data of 29 countries. They constructed a model for the expectation of income growth, inflation, interest rates, and current accounts, and their results showed that the fundamental factor expectations were more important in the long run than in the short run. Furthermore, they found that the expectation of an increase in domestic income in comparison to income of the USA would result in an unexpected appreciation of the local currency. On the other hand, they found that expectations of an inflation increase would result in an expected depreciation. Furthermore, [6] used a panel analysis method to assess the movement of the real exchange rate in 15 emerging countries. Their results showed the commodity prices tended to be an important factor that determined the real exchange rate movements in those countries. However, the central banks in those countries have moderately stabilized exchange rates. Additionally, the money supply growth in those countries was found to have a positive effect on the real exchange rate, whereas an increase in the money supply was found to result in the appreciation of the real exchange rate.

The impact of the money supply on the exchange rate can be observed via inflation. Ref. [7] investigated the relationship between inflation and the real exchange rate for the Pakistani Rupee. They applied the Johansen cointegration method to test the variable relationships in their model, and their results showed that there were long-run relationships among the variables. They also found that an increase in the money supply led to a decrease in inflation and an increase in the exchange rate. These results are consistent with the idea that an inadequate money supply may have an inverse relationship with prices as long as the growth of the money supply is consistent with the growth of the economy. Ref. [8] analyzed the Bangladesh exchange rate of the Taka to the USD with respect to an econometric model that included the money supply, inflation, and current accounts. The cointegration test proved the existence of a long-run relationship and that the money supply had a positive impact on the exchange rate. An increase in the money supply resulted in the Taka depreciating against the USD. However, money supply cannot determine the Granger causality of the exchange rate; only when all the model variables are summed will the null hypothesis of no Granger causality be rejected. The exchange rate of the local currency for the USD does not accurately reflect real exchange rate fluctuations. The real effective exchange rate (REER) has been widely used in real exchange rate analysis. [9] implemented a cointegration method to analyze the impact of the money supply, trade balance, and inflation on the Kenyan REER; their cointegration test results confirmed the existence of long-run relationships among the variables. There was also a positive relationship between the variables and the REER; this positive relationship indicates that an increase in the money supply will depreciate the real exchange rate and also increase inflation.

Natural resources are classified as major sources of income in developing countries, including Saudi Arabia. Ref. [10] used structural vector autoregression (SVAR) to distinguish between the impact of oil on the exchange rate for MENA oil-importer and oil-exporter countries. The results showed that the real exchange rate fluctuations could mainly be explained by oil prices in Tunisia and Morocco, which are oil-importer countries. The oil price was found to explain the fluctuations in both the real exchange rate and inflation in Bahrain and Saudi Arabia, which are oil-exporter countries. Furthermore, Iran's inflation fluctuations were shown to be explained by the oil price. However, unlike the rest of the

studied countries, oil could not explain the fluctuations in both the real exchange rate and inflation in Algeria. Ref. [11] examined the relationship between the REER and total productivity in high-income and upper-middle-income countries using an econometric GMM method. Their results showed that in high-income countries, productivity increases caused the REER to depreciate, whereas in upper-middle-income countries, productivity increases caused the REER to appreciate. Furthermore, financial development and natural resource rents were found to have significant impacts on the REER in high-income countries compared with upper-middle-income countries, in which the effects of those factors were shown to be insignificant. Additionally, trade openness was shown to have a significant impact on the REER variations in both types of countries. Ref. [12] used an ARDL cointegration framework to assess the relationship between oil and the REER in Saudi Arabia, and their econometric estimations confirmed long-run relationships in their model; specifically, their results suggested a direct causal relationship between oil and the REER in the short run and a bidirectional causal relationship in the long run.

Government expenditure is a major macroeconomic factor that plays a significant role in a country's economy. Ref. [13] applied ARDL cointegration to assess the relationships among productivity differences, government expenditure, foreign investment, trade openness, interest rate differentials, inflation differentials, terms of trade, foreign reserves, and net foreign assets to determine the real exchange rate of India; their results proved the existence of long-run relationships among the studied variables. However, [14] argued that the impacts of fiscal and monetary policies depend on the fiscal regime. Furthermore, they found that the effects of contractionary monetary policies on the real exchange rate were dominated by expansionary fiscal policies, as the real exchange rate was shown to depreciate as an outcome of contradictory polices. For Brazil, such an outcome could be avoided if the government offset the debt accrued due to the increase in its current budget with a future budget surplus. Ref. [15] analyzed the impact of disaggregated government spending in Latin American countries. Their results showed that government consumption depreciated the real exchange rate in contrast to government investment, which appreciated the real exchange rate. Ref. [16] estimated the VAR for 38 countries to examine the impact of government spending on the real exchange rate for different exchange rate regimes. They found that in countries with fixed exchange rates, government spending exhibited asymmetric effects on the real exchange rate. Their results showed that increases in government spending, in contrast to increases in economic output and employment, caused the real exchange rate to appreciate. Ref. [17] assessed the impact of government spending on the real exchange rate in Ethiopia. Using VAR analysis, they showed that increases in government spending caused Ethiopia's real exchange rate to appreciate, while government investment had no impact on the real exchange rate.

Economic sustainability is the focus of governments that intend to wisely utilize their finite economic resources to enhance their economies. Ref. [18] analyzed the impact of companies' application of social responsibility concepts on the exchange rate of the USD for major world currencies. They used a generalized autoregressive conditional heteroskedasticity (GARCH) model to test for the volatility spillover from the stock returns of CSR companies to the USD exchange rate return against major world currencies. Their results revealed that the impact of CSR companies on the USD exchange rate against major world currencies was negative. Furthermore, they found that increases in the Dow Jones Sustainability World Index (DJSWI), which is used to measure the sustainability of companies, leads such companies to transfer their profits outside the USA, therefore depreciating the USD against major world currencies. On the other hand, a decrease in the DJSWI would indicate that the global business environment is unstable, causing investors to transfer their wealth to the USA's secure business environment and therefore appreciating the USD against major world currencies. However, [19] assessed the performance of environmental, social, and governance (ESG) approaches in predicting future exchange rate fluctuations; the study used data for 42 countries listed in the MSCI ESG index. The results indicated that the performance of a country in the ESG index was significantly related to its exchange

rate performance, and countries with high ESG index ratings showed improved currency performance. Ref. [20] used a hybrid machine learning method to examine the impact of macroeconomic policy news on exchange rate fluctuations. Using subjective information, the model was used to test the ability of such an approach to predict future exchange rates in the long term, and the model indicated the significant impact of subjective information on exchange rate fluctuations. The model also revealed a strong correlation between the publication of macroeconomic news and the daily exchange rate fluctuations, as well as the reliability of the model in predicting short-run exchange rates. On the other hand, the long-run exchange rate stability was found to rely on the sustainability of macroeconomic policies. These countries have the responsibility to ensure the implementation of the sustainable programs established by world organizations. Ref. [21] assessed the environment and moderation policies considered by central banks across the world. They distinguished between the impact of environmental factors on the classic goals of central banks and the possibility of the central banks financing green and sustainable projects in order to assess which environmental policies should be mandated by the government. Out of the 133 assessed central banks, 12% had an explicit sustainability mandate, and 29% had sustainability goals to urgently implement the mandated government policies. Additionally, they reported that central banks without explicit or implicit sustainability polices will incorporate global sustainability policies to ensure the stability of their macroeconomics and, therefore, the stability of their assets and the exchange rate.

To summarize the literature, the macroeconomics fundamental factors such as income, consumer price index, money supply, natural resources and government expenditure have an impact on the real exchange rate. The literature on the exchange rate and sustainable economic development lacks studies covering a wide range of economies with different economic structures. This study is intended to fill this research gap by considering the impact of sustainable economic development on the real exchange rate of Saudi Arabia, which is a G20 member and the leader of the Organization of the Petroleum Exporting Countries (OPEC). The size and structure of Saudi Arabia's economy mean that our study's results will be applicable to understanding the role of sustainable economic development on the real exchange rate in one of the emerging countries with a significant role in the world by its leading oil producers' countries. This study will open the arena of studies focusing on sustainability in developing countries that rely on natural resources with high carbon emissions as the main source of income. Those studies will help in understanding the challenges that the economy may face in the transition toward sustainability. However, this study, by focusing on Saudi Arabia, may have a different emphasis than the rest of the studies in sustainability that are mainly focusing on developed economies.

## 3. Methodology

### 3.1. Theoretical Model

Saudi Arabia has pegged its currency, the Saudi riyal, to the USD since 1986, based on the Saudi Arabia Central Bank (SAMA), and the relevant exchange rate modeling is similar to that applied to both fixed exchange rate regimes and flexible exchange rate regimes, following the work of [22,23]. In this study, we applied the Monetary Approach to assess the real exchange rate's determinants. This Monetary Approach depends on the Purchasing Power Parity (PPP) theory and the Quantity Theory of Money (QTM). These two theories characterize the exchange rate as the relative prices between two economies. Furthermore, the interest rate is an exogenous determinant of the exchange rate in the long run since the interest rate is determined by global markets. The Monetary Approach relies on the assumption that PPP holds over time, which means that prices are instantly adjusted in global markets to achieve equilibrium. [24,25] found that the PPP held steady over multiple half-lives, and [26] found that the PPP could hold for more than its estimated half-life during recent periods of globalization. The PPP is used to model the exchange rate as

relative prices, where Equation (1) expresses the exchange rate as the difference between the prices in a domestic economy and a foreign economy.

$$S_t = P_t - P_t^F + c + \varepsilon_t \quad \begin{array}{l} \text{If } c = 0, \text{ there is an absolute PPP relationship.} \\ \text{If } c \neq 0, \text{ there is relative PPP} \end{array} \quad (1)$$

where $S_t$ is the exchange rate in logarithmic form; $P_t$ and $P_t^F$ are the prices in the local and foreign economies, respectively; c is a constant; and $\varepsilon_t$ is an error term indicating the disturbance of the estimation. Equation (1) shows that the exchange rate appreciates as domestic prices increase. On the other hand, the exchange rate depreciates when foreign prices increase. Remember that $S_t$ is a local currency unit that needs to match one unit of a foreign country's currency. The Monetary Approach relies on the QTM as an explanation of the exchange rate behavior, and the QTM depends on the monetary policies in both countries because those policies can explain the exchange behaviors via the quantity of money in both economies. Furthermore, the local and foreign demands for money are functions of income, price level, and the nominal interest rate. These relationships can be expressed as Equations (2) and (3), respectively.

$$m_t = p_t + \beta_1 y_t - \beta_2 i_t + \omega_t \quad (2)$$

$$m_t^* = p_t^* + \beta_1^* y_t^* - \beta_2^* i_t^* + \omega_t^* \quad (3)$$

where $m_t$ and $m_t^*$ are the local and foreign demands for money, respectively; $y_t$ and $y_t^*$ are the local and foreign levels of income, respectively; $i_t$ and $i_t^*$ are the local and foreign interest rates, respectively. In general, the demand for money is positively affected by the price level and income and negatively affected by the interest rates. Solving Equations (2) and (3) results in Equations (4) and (5), respectively.

$$p_t = m_t - \beta_1 y_t + \beta_2 i_t - \omega_t \quad (4)$$

$$p_t^* = m_t^* - \beta_1^* y_t^* + \beta_2^* i_t^* - \omega_t^* \quad (5)$$

where $\beta_1$ and $\beta_1^*$ represent the local and foreign elasticity in the money demand with respect to income, respectively; $\beta_2$ and $\beta_2^*$ represent the local and foreign semi-elasticity in the money demand with respect to interest rates, respectively; $\omega_t$ and $\omega_t^*$ are error terms representing the disturbance components of the estimation.

From Equations (4) and (5), we can proceed to Equation (6) by substituting the price level in Equation (1).

$$s_t = m_t - m_t^* - \beta_1 y_t + \beta_1^* y_t^* + \beta_2 i_t - \beta_2^* i_t^* + c + \varepsilon_t^*. \quad (6)$$

Equation (6) does not consider the economic structure, so $\beta_1 \neq \beta_1^*$. Moreover, interest rates rely on the international financial markets, and the failure of those markets is a possible outcome when Equation (6) is used to uncover the interest parity; the uncovered interest parity suggests that a currency with a high interest rate is expected to depreciate relative to a currency with a low interest rate. We can express this idea with Equation (7).

$$E_t(\Delta s_{t+k}) = (i_t^* - i_t) + \vartheta \quad (7)$$

where $\Delta s_{t+k}$ is the expected change in the exchange rate, which is positively affected by increases in the foreign interest rate relative to the local interest rate, and $\vartheta$ represents the risk premium of holding local assets. We can substitute $E_t(\Delta p_{t+k} - \Delta p_{t+k}^*)$ into Equation (7) to create the following equation:

$$q_t = E_t(q_{t+k}) + (r_t - r_t^*) + \theta_t \quad (8)$$

where $r_t = i_t − E_t(\Delta p_{t+k})$, which represents the expected real interest rate in the home country, and $r_t^* = i_t^* − E_t(\Delta p_{t+k}^*)$, which represents the expected real interest rate in the foreign country. Furthermore, $q_t = s_t + p_t − p_t^*$, which represents the expected real exchange rate and $\theta_t$ is the disturbance term of the equation. Equation (8) has two apparatuses for determining the real exchange rate: the expectation of the real exchange rate at time $t + k$ and the differentiation of the interest rate at maturity that is accrued at $t + k$. However, one must assume unobservable variables when calculating the exchange rate expectations based on the fundamental elements that impact an economic equilibrium. $\bar{q}$ represents the fundamentals affecting the equilibrium of the exchange rate, which can be used to rewrite Equation (8) as follows, with $\dot{q}$ representing the exchange rate at equilibrium.

$$\dot{q}_t = \bar{q}_t + (r_t − r_t^*) \tag{9}$$

where $\bar{q}_t$ represents the fundamental variables (in contrast to theoretical variables) in the economy that impact the exchange rate, thus reflecting the behavior of the exchange rate.

The theoretical frame has addressed the importance of the factors in the model. These factors will be incorporated in the empirical framework to deliver an insight into the impact of independent factors on the dependent factor. In this case, the dependent factor is the real exchange rate, and the aim is to assess the effect of each independent factor on the real exchange rate as a dependent factor.

### 3.2. Empirical Model

An empirical model has to capture the impact of its independent variables on its dependent variables. In our study, we considered using two models to achieve our objectives: the Johansen cointegration [23] and ARDL cointegration models. The choice between these models depended on the stationarity test of the variables. If the variables were found to be stationary at rank one or zero but not two, then the [23] method would be appropriate for this study. However, if at least one of the variables were at a different rank in its stationarity test, then ARDL cointegration would be a suitable method for this study. To build our econometric model, we considered which function could be used to test which variables could impact the real exchange rate. We noted that Equations (6) and (9) had variables that impact the real exchange rate. Equation (6) revealed that the money supply, income, and interest rate differences had impacts on the real exchange rate. However, Saudi Arabia pegs its exchange rate to the USD, which weakens the impact of world interest rate changes on the Saudi riyal. A country that pegs its exchange to another is only affected by the changes in that other country; see the work of [27]. Accordingly, the world interest rate has no impact on the exchange rate in Saudi Arabia. Equation (9) indicates that economic activity indicators are wide-ranging, and this study focused on the Sustainable Development Index; our control factors were set as government expenditure and oil rent, and we did not consider the terms of trade and net foreign assets. The Gulf Cooperation Council countries' terms of trade are insignificant due to their dependence on oil exportation; see the work of [28]. Furthermore, oil rent plays the same role as oil prices since oil rent accounts for the difference between the revenue and the cost of oil. Additionally, Saudi Arabia's foreign assets have positive correlations with oil prices, so increases in the oil price will lead to increases in foreign asset accumulation; see the work of [29]. Accordingly, involving net assets in the model resulted in multicollinearity, and oil rent was used to assess the impact of the net foreign assets. Therefore, the real exchange rate function can be written as follows:

$$LREER = f(LY,\ LP, LM2,\ LSDI,\ LOIR,\ LEX) \tag{10}$$

Equation (10) shows that the real exchange rate is a function of income, price, money supply, sustainability, oil rent, and government expenditure.

The variables in Equation (10) are as follows: LREER is the real effective exchange rate in logarithmic form, LY is the income differential in logarithmic form, LP is the price level differential in logarithmic form, LM2 is the money supply differential in logarithmic form,

LSDI is the Sustainability Development Index in logarithmic form, LOIR is the oil rent in logarithmic form, and LEX is the government expenditure as a percentage of the GDP in logarithmic form.

The data description and stationarity tests were conducted before choosing the econometric method.

### 3.3. Data

Our dataset was acquired from different databases covering the period between 1990 and 2019. The world real GDP, world money supply, and world CPI data were acquired from the World Bank database. The Saudi real GDP, Saudi CPI, and Saudi government expenditure as percentages of GDP were acquired from the World Outlook Database, which is hosted by the International Monetary Fund (IMF). The real effective exchange rate (REER) data were acquired from the International Financial Statistics (IFS) database, which is hosted by the IMF. Finally, the Sustainable Development Index (SDI) data were acquired from Jason Hickel's website based on his [30] study (https://www.sustainabledevelopmentindex.org/ accessed on 30 April 2023).

#### 3.3.1. Data Description

The data's descriptive statistics showed that all the variables except the CPI were normally distributed, and the standard deviation for all the variables did not exceed 0.5. Furthermore, the kurtosis for all the variables except government expenditure and CPI did not exceed 2. Table 1 shows the variables' statistical information.

**Table 1.** Descriptive statistics.

|  | LREER | LY | LP | LM2 | LSDI | LOIR | LEX |
|---|---|---|---|---|---|---|---|
| Mean | 4.7210 | −5.1746 | 0.1623 | −5.8854 | 3.0396 | 3.5296 | 3.5202 |
| Median | 4.7399 | −5.0602 | 0.0142 | −5.9521 | 3.1711 | 3.5088 | 3.5185 |
| Maximum | 4.8579 | −4.5885 | 0.7357 | −5.3545 | 3.4544 | 3.9905 | 3.7175 |
| Minimum | 4.5465 | −6.0033 | −0.0608 | −6.3653 | 2.4595 | 2.9959 | 3.2839 |
| Std. Dev. | 0.0992 | 0.4726 | 0.2452 | 0.3383 | 0.3424 | 0.2807 | 0.1073 |
| Skewness | −0.2798 | −0.3377 | 1.1623 | 0.2118 | −0.7970 | −0.0819 | −0.3071 |
| Kurtosis | 1.6571 | 1.6305 | 2.9367 | 1.5291 | 1.9722 | 1.8998 | 2.8649 |
| Jarque–Bera | 2.6458 | 2.9146 | 6.7597 | 2.9284 | 4.4967 | 1.5463 | 0.4944 |
| Probability | 0.2663 | 0.2328 | 0.0340 | 0.2312 | 0.1055 | 0.4615 | 0.7809 |
| Observations | 30 | 30 | 30 | 30 | 30 | 30 | 30 |

Table 2 shows the descriptive statistics of the variables in their first difference. The standard deviation was within the range of 0.03 to 0.13, except for that of oil rent, which exceeded 0.20 due to oil price fluctuations. Except for government expenditure and the Sustainability Development Index, the variables were normally distributed in their first difference.

**Table 2.** Descriptive statistics for the first difference.

|  | DLREER | DLY | DLP | DLM2 | DLSDI | DLOIR | DLEX |
|---|---|---|---|---|---|---|---|
| Mean | −0.0041 | 0.0487 | −0.0274 | 0.0277 | −0.0236 | −0.0222 | −0.0053 |
| Median | −0.0082 | 0.0384 | −0.0300 | 0.0251 | −0.0042 | −0.0357 | −0.0295 |
| Maximum | 0.0900 | 0.1879 | 0.0471 | 0.1015 | 0.2196 | 0.4293 | 0.3287 |
| Minimum | −0.0880 | −0.1022 | −0.0869 | −0.0741 | −0.6019 | −0.5206 | −0.1462 |
| Std. Dev. | 0.0409 | 0.0732 | 0.0318 | 0.0563 | 0.1300 | 0.2237 | 0.1055 |
| Skewness | 0.1612 | −0.0742 | 0.2396 | −0.2728 | −2.8959 | −0.4526 | 1.1155 |
| Kurtosis | 2.7471 | 2.3380 | 2.5181 | 1.8118 | 15.0932 | 2.8843 | 4.5271 |
| Jarque–Bera | 0.2028 | 0.5560 | 0.5581 | 2.0655 | 217.250 | 1.0064 | 8.8335 |
| Probability | 0.9035 | 0.7572 | 0.7564 | 0.3560 | 0.0000 | 0.6045 | 0.0120 |
| Observations | 29 | 29 | 29 | 29 | 29 | 29 | 29 |

### 3.3.2. Unit Root Test

We used the augmented Dickey–Fuller (ADF) unit root test to assess the null hypothesis that the series had unit roots in opposition to the hypothesis that the series did not have unit roots. Additionally, the Kwiatkowski–Phillips–Schmidt–Shin (KPSS) test, which has the opposite null hypothesis to the ADF, was performed to ensure that the series' unit root results were robust.

Table 3 shows the unit root test variables. According to the ADF test, all the variables except the price differential and government expenditure exhibited unit roots. However, the KPPS test showed that oil rent and government expenditure were stationary, and the rest of the variables exhibited unit roots.

**Table 3.** Unit root test results.

| Variable | ADF $H_O$: Variable Has a Unit Root Level | | KPSS $H_O$: Variable Is Stationary Level | |
|---|---|---|---|---|
| | Intercept | Intercept and Trend | Intercept | Intercept and Trend |
| LREER | −1.833171 (0.3575) | −1.601651 (0.7665) | 0.535649 ** (0.038) | 0.157603 ** (0.044) |
| LY | −1.418547 (0.5594) | −1.402192 (0.8388) | 1.06602 *** ($p < 0.01$) | 0.210706 *** ($p < 0.01$) |
| LP | −5.447411 *** (0.0001) | −1.868833 (0.6445) | 0.885533 *** ($p < 0.01$) | 0.273902 *** ($p < 0.01$) |
| LM2 | −0.337146 (0.9073) | −3.288705 * (0.0888) | 1.03691 *** ($p < 0.01$) | 0.150167 ** (0.049) |
| LSDI | −0.453929 (0.8865) | −2.786892 (0.2129) | 0.911462 *** ($p < 0.01$) | 0.215494 *** ($p < 0.01$) |
| LOIR | −2.242743 (0.1965) | −2.204331 (0.4697) | 0.173146 ($p < 0.10$) | 0.173148 ** (0.034) |
| LEX | −3.159634 ** (0.0331) | −3.056226 (0.1352) | 0.189128 ($p < 0.10$) | 0.154871 ** (0.046) |

*, **, and *** denote statistical significance at the 10%, 5%, and 1% levels, respectively. *p*-values are shown in parentheses.

Table 4 shows the unit root test results of all variables in their first difference. The ADF test showed that all the variables except for the price differential were stationary in their first difference. The KPPS test results, including the trend, also showed that all the variables except for the price differential were stationary in their first difference.

**Table 4.** Unit root test results for the first difference.

| Variable | ADF $H_O$: Variable Has a Unit Root First Difference | | KPSS $H_O$: Variable Is Stationary First Difference | |
|---|---|---|---|---|
| | Intercept | Intercept and Trend | Intercept | Intercept and Trend |
| LREER | −3.790226 *** (0.0079) | −3.866583 ** (0.0275) | 0.177402 ($p > 0.10$) | 0.0922687 ($p > 0.10$) |
| LY | −4.716009 *** (0.0008) | −3.905869 ** (0.0271) | 0.205311 ($p > 0.10$) | 0.0625668 ($p > 0.10$) |
| LP | −1.454300 (0.5402) | −0.167473 (0.9903) | 0.768447 ($p > 0.10$) | 0.174299 ** (0.033) |
| LM2 | −3.863939 *** (0.0066) | −3.824522 ** (0.0301) | 0.129688 ($p > 0.10$) | 0.125693 (0.094) |
| LSDI | −6.490189 *** (0.0000) | −6.471555 *** (0.0001) | 0.200072 ($p > 0.10$) | 0.0718062 ($p > 0.10$) |

**Table 4.** *Cont.*

| Variable | ADF $H_O$: Variable Has a Unit Root First Difference | | KPSS $H_O$: Variable Is Stationary First Difference | |
|---|---|---|---|---|
| | Intercept | Intercept and Trend | Intercept | Intercept and Trend |
| LOIR | −5.403681 *** (0.0001) | −5.404521 *** (0.0008) | 0.109453 ($p > 0.10$) | 0.107678 ($p > 0.10$) |
| LEX | −6.551915 *** (0.0000) | −6.426239 *** (0.0001) | 0.0902991 ($p > 0.10$) | 0.0499627 ($p > 0.10$) |

\*\*, and \*\*\* denote statistical significance at the 5%, and 1% levels, respectively. *p*-values are shown in parentheses.

Tables 3 and 4 show that at least one variable was stationary at its level, as proved by the ADF and KPPS tests: government expenditure. Therefore, we chose ARDL cointegration as the econometric methodology for our economic model. Furthermore, all the variables were either I(0) or I(1), and no variable was ranked I(2), which fulfilled the ARDL cointegration requirements.

### 3.4. ARDL Cointegration Method

ARDL cointegration is used to examine the existence of long-run relationships among key variables. ARDL cointegration can be used to assess the dynamic relationships among variables to correct the misalignment of a model due to equilibrium. This method has advantages compared with Johansen's cointegration method, such as the ability to estimate a model with different lags among variables without the same rank order. Furthermore, ARDL cointegration can be used to perform estimation for models with lower frequency datasets than those of Johansen's cointegration method because ARDL cointegration uses the Bond test instead of the regularly used t- and F-tests. Moreover, ARDL cointegration uses dependent and independent variables, which solves the exogeneity and endogeneity issues of the variables, see the work of [31,32]. Furthermore, the method can be used to clarify autocorrelation with lag settings, and tests for heterogeneity and normality were conducted to ensure the model's reliability.

The model was estimated based on Equation (10); we transformed Equation (10) into the following equation to set the ARDL cointegration:

$$LEER_t = \alpha + \beta_1 LY_t + \beta_2\ LM2_t + \beta_3\ LP_t + \beta_4\ LSDI_t + \beta_5 LOIR_t + \beta_6\ LEX_t + \varepsilon_t \quad (11)$$

The model was estimated with Equation (11) as the first step towards ARDL cointegration.

For cointegration, Equation (11) was transformed into the ARDL cointegration form as follows:

$$\Delta LREER_t = \alpha + \sum_{i=1}^{p} \beta_i \Delta LREER_{t-i} + \sum_{i=0}^{q} \delta_i \Delta LY_{t-i} + \sum_{i=0}^{r} \gamma_i \Delta LM2_{t-i} +$$
$$\sum_{i=0}^{k} \partial_i \Delta LP_{t-i} + \sum_{i=0}^{d} \rho_i \Delta LSDI_{t-i} + \sum_{i=0}^{g} \sigma_i \Delta LOIR_{t-i} + \sum_{i=0}^{w} \omega_i \Delta LEX_{t-i} +$$
$$\theta_0 LREER_{t-1} + \theta_1 LY_{t-1} + \theta_2 LP_{t-1} + \theta_3 LM2_{t-1} + \theta_4 LSDI_{t-1} + \theta_5 LOIR_{t-1} +$$
$$\theta_6 LEX_{t-1} + \varepsilon_t \quad (12)$$

where $\beta$, $\delta$, $\gamma$, $\partial$, $\rho$, $\sigma$, and $\omega$ are the short-run coefficients, and $\theta$s is a long-run coefficient based on the work of [33]. Furthermore, $\theta$s was used to assess the instant impact of the independent variables on the dependent variables, while $\beta$, $\delta$, $\gamma$, $\partial$, $\rho$, $\sigma$, and $\omega$ were used to examine the long-run impacts and speed adjustments toward equilibrium from the short-run dynamics. $\beta$, $\delta$, $\gamma$, $\partial$, $\rho$, $\sigma$, and $\omega$ were determined based on the information criteria lag-selection methods (e.g., the AIC, SC, and BIC).

Equation (12) was used to examine the long-run relationships among the variables that were set based on Equation (10) and transformed into Equation (12). Therefore, the null hypothesis and alternative hypothesis were clarified to be able to examine the relationships

among the variables. Ref. [34]'s *t*-test and [33]'s bound F-statistic were applied to the regressors in the model to either accept or reject the null hypothesis. The null hypothesis and alternative hypothesis for the model were as follows, respectively:

$$H_0 : no\ cointegration\ exists.$$

$$H_1 : cointegration\ exists.$$

The F-test was applied to the joint coefficients with different lag selections to accept or reject the null hypothesis based on the following set:

$$H_0 :\ \theta_0 = \theta_1 = \theta_2 = \theta_3 = \theta_4 = \theta_5 = \theta_6 = 0 ;$$

$$H_1 :\ \theta_0 \neq 0,\ \theta_1 \neq 0,\ \theta_2 \neq 0,\ \theta_3 \neq 0,\ \theta_4 \neq 0,\ \theta_5 \neq 0,\ \theta_6 \neq 0$$

Ref. [35]'s F-test bound table, which is used for joint coefficient estimation with low data frequency, was used as a robustness check since the data span was relatively small. [36] also provided a *t*-test that can be used to ensure that the lag coefficient of a dependent variable is significant. The hypotheses of the *t*-test, focusing on the coefficient of the lag of the dependent variable, were as follows:

$$H_0 :\ \theta_0 = 0;$$

$$H_1 :\ \theta_0 < 0.$$

The rule for the F-test is that the null hypothesis is rejected if the estimation's F-statistic is greater than I(1) in the [33]'s F-statistic bound table; at this point, the variables have long-run relationships. In this study, the F-statistic was tested against [35]'s bound table by applying the same rule. If the F-statistic for the estimation was lower than I(0), then the variables did not have long-run relationships, and a short-run dynamic relationship existed. Furthermore, if the estimation's F-statistic was greater than I(1), then [35]' *t*-test was performed for the lag coefficient of the dependent variable with the same F-statistic rule. The null hypothesis of the t-statistic was rejected for coefficients that were lower than the t-statistic in [35]'s table; otherwise, the model's variables demonstrated long-run relationships. During cross-checking, the lag coefficient of the dependent variable should be negative. However, if the test result is between the upper and lower bounds, the relationships among variables are ambiguous.

If the null hypotheses were not rejected, short-run dynamic estimation with speed adjustment was performed. The error correction mechanism (ECM) was used to ensure that the model converged toward equilibrium in the long run in case any distortion occurred in the short run. The estimation of the short-run dynamics is represented by the following equation:

$$\Delta LERRE_t = \alpha + \sum_{i=1}^{p} \beta_i \Delta LERRE_{t-i} + \sum_{i=0}^{q} \delta_i \Delta LY_{t-i} + \sum_{i=0}^{r} \gamma_i \Delta LM2_{t-i} + \\ \sum_{i=0}^{k} \partial_i LP_{t-i} + \sum_{i=0}^{d} \rho_i LSDI_{t-i} + \sum_{i=0}^{g} \sigma_i \Delta LOIR_{t-i} + \sum_{i=0}^{w} \omega_i \Delta LEX_{t-i} + \\ \varphi ECM_{t-i} + \varepsilon_t \tag{13}$$

The variables in Equation (13) are in their first difference, representing the short-run dynamics. The most important coefficient in this equation is $\varphi$, which represents the speed of the adjustment of the model toward equilibrium in case short-run shocks occur. This speed adjustment connects the variables across their two-time horizons, the long- and short-run dynamics. This coefficient should be negative and significant, as a negative sign ensures convergence toward equilibrium.

ARDL cointegration requires Equation (11) to have no serial correlations and be normally distributed, so the LM test was performed to ensure that the model did not exhibit serial correlations, and the Jarque–Bera test was applied to ensure that the model was normally distributed. Furthermore, the Breusch–Pagan–Godfrey test was used to assess the model's heteroscedasticity. CUSUM and CUSUM squares were applied to ensure

the model's stability. These diagnostic tests were applied to ensure the reliability of the model based on the work of [16,37].

## 4. Results

### 4.1. Primary Estimation

The estimation of ARDL cointegration began with the estimation of the primary equation, and the lag size was determined based on the lag criteria before the estimation. Here, there were less data, so the Akaike information criterion (AIC) was used instead of the [38] criterion. The optimal lag for the model based on the AIC was two lags, with a test value of $-20.63922$.

The ARDL model was estimated with Equation (11); all the estimation coefficients are shown in Table 5. Table 5 shows that the dependent variable was the LREER, and the rest of the variables were independent. The AIC showed that the optimal lag length for the variables was $p = 1$, $q = 0$, $r = 0$, $k = 2$, $d = 1$, $g = 2$, and $w = 2$. This trend was significant at a 1% probability. Model diagnostics were run to ensure that the model did not exhibit serial correlation and was normally distributed, and the model did not exhibit heteroscedasticity.

**Table 5.** Primary ARDL cointegration model estimations.

| Variable | Coefficient | *t*-Statistic | Probability |
|---|---|---|---|
| LREER(−1) | 0.409399 *** | 4.497264 | 0.0007 |
| LY | 0.543635 *** | 4.500687 | 0.0007 |
| LM2 | 0.034347 | 0.336770 | 0.7421 |
| LP | 0.462469 | 1.741939 | 0.1071 |
| LP(−1) | −0.191086 | −0.597969 | 0.5610 |
| LP(−2) | −0.439512 | −1.361771 | 0.1983 |
| LSDI | −0.048815 | −1.503060 | 0.1587 |
| LSDI(−1) | −0.134259 *** | −3.492202 | 0.0044 |
| LOIR | −0.309840 *** | −7.214559 | 0.0000 |
| LOIR(−1) | −0.032019 | −1.009956 | 0.3324 |
| LOIR(−2) | −0.148308 *** | −5.450133 | 0.0001 |
| LEX | 0.205782 *** | 3.840663 | 0.0023 |
| LEX(−1) | 0.025923 | 0.436671 | 0.6701 |
| LEX(−2) | −0.156707 ** | −2.807979 | 0.0158 |
| C | 8.529832 *** | 7.663873 | 0.0000 |
| @TREND | −0.044433 *** | −6.391937 | 0.0000 |

** and *** denote statistical significance at the 5%, and 1% levels, respectively.

### 4.2. Diagnostic Tests

Table 6 shows that the model did not exhibit serial correlations among the independent variables, thus failing to reject the null hypothesis that there would be no serial correlations for up to two lags. There was no heteroscedasticity in the model, as the test failed to reject the null hypothesis of no heteroscedasticity. Furthermore, the Jarque–Bera test showed that the model was normally distributed. Thus, the model showed no econometric prejudices in the time-series data. The model was then tested for stability to ensure there were no structural breaks that would make the results unrealistic, see the work of [37].

**Table 6.** Model diagnostic test results.

| Test | $\chi^2$ | Probability |
|---|---|---|
| Breusch–Godfrey Serial Correlation LM Test | 2.190410 | 0.3345 |
| Breusch–Pagan–Godfrey Heteroscedasticity Test | 1.314487 | 0.3205 |
| Jarque–Bera Test | 0.071473 | 0.9648 |

Figure 1 shows that the CUSUM plot was within the 5% boundary, which means that the model had no structural breaks and was stable.

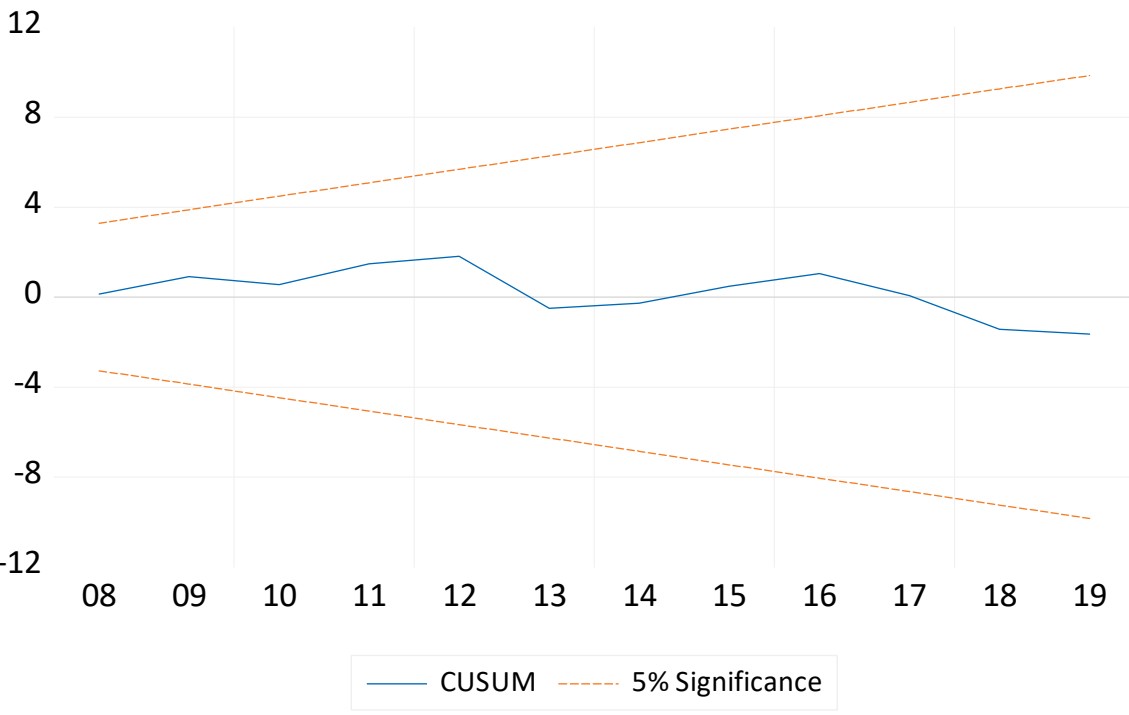

**Figure 1.** Plot of the CUSUM test.

Figure 2 shows that the CUSUM squares plot was within the 5% boundary, which confirmed there were no structure breaks, and the model was stable. These robustness checks ensured the stability of the model.

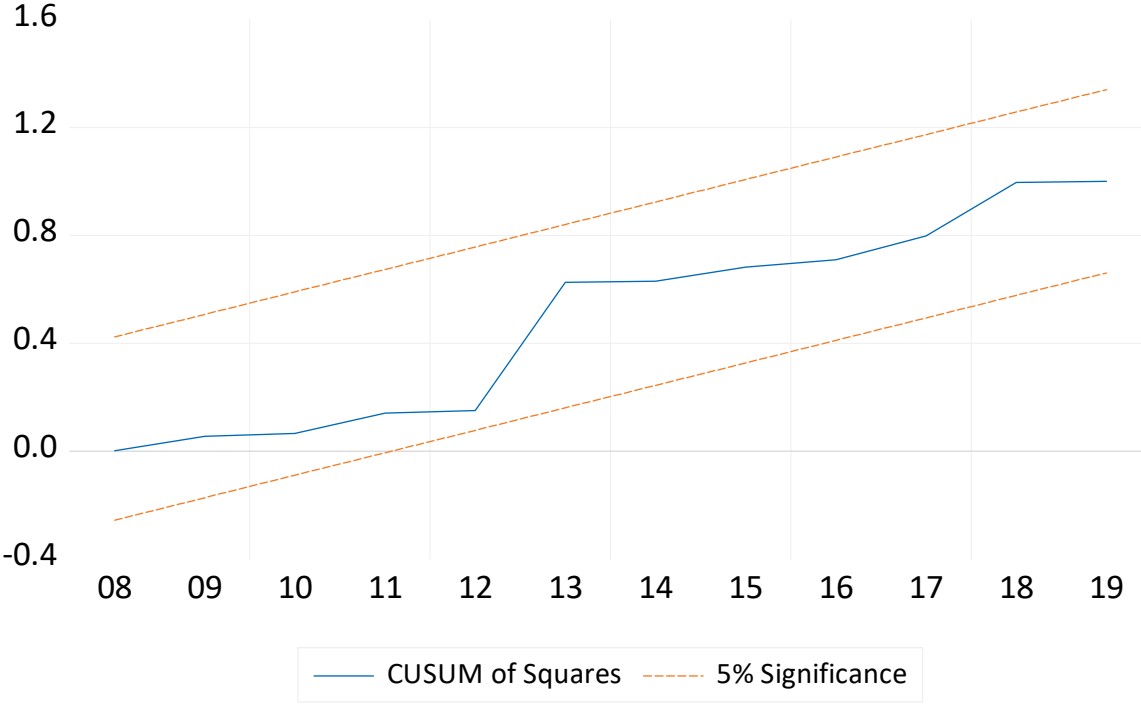

**Figure 2.** Plot of the CUSUMSQ test.

The model's R-squared was 0.9899, which would mean that almost 99% of the variations in the dependent variables could be explained by the model, and the rest could be explained by the error term. Then, the adjusted R-squared value was 0.9772, which means that most of the variations in the dependent variables were explained by the model, and about 2% were explained by the error term. The Durbin–Watson test was used to show that the model did not have first-order autocorrelation by comparing the model's statistics and the critical DW statistic. The DW statistic in the model was 2.3113, which exceeded the critical DW statistic of 1.729 for k = 5 and n = 28. Furthermore, the F-statistic for the model was 78.4932, with a p-value of 0.0000, which rejected the null hypothesis that the coefficients would be zero. Hence, the ARDL cointegration was robust according to all statistical tests, which indicates that the model was statistically sound.

*4.3. ARDL Cointegration*

ARDL cointegration was performed to determine whether there were long-run relationships between the variables in the model based on the work of [33,35] in tandem with the critical F-statistic value. The F-statistic for the model was 11.78708, with a lag distribution of (1, 0, 0, 2, 1, 2, 2). A long-run relationship was known to exist if the F-statistic in the model was greater than the critical upper bound F-statistic. Therefore, the estimation relied on Case V in [33], in which k = 6 (see Table CI(v) p.301). The estimation was also based on Case V of [35] (p. 1990), in which k = 6 and n = 30. However, the estimation was based on fewer observations than those in [35] table so the value closest to the critical value was used. The critical F-statistic values of [33,35] for k = 6 are shown in Table 7. The estimated F-statistic exceeded the I(1) value at the 1% significance level; therefore, the null hypothesis was rejected, and the alternative was accepted. The rejection of the null hypothesis indicated there were long-run relationships among the model's variables. Next, the $LREER_{t-1}$ coefficient was tested based on [33] to ensure the significance of the coefficient within the bound test. The t-statistic of the coefficient was −6.4877, which was greater than I(1) at the 1% level of significance. The critical values for 1%, 5%, and 10% at k = 6 were expressed with the lower bound I(0) and the upper bound I(1) (−3.96, −5.31), (−3.41, −4.69), and (−3.13, −4.37), respectively. The test indicated the rejection of the null hypothesis that the dependent lag coefficient would be zero, so we accepted the alternative hypothesis that the dependent lag would be less than zero.

**Table 7.** Critical Pesaran and Narayan bound test values.

| Critical Values | Pesaran | | Narayan | |
| --- | --- | --- | --- | --- |
| | Lower Bound I(0) | Upper Bound I(1) | Lower Bound I(0) | Upper Bound I(1) |
| 1% | 3.60 | 4.90 | 5.046 | 6.930 |
| 5% | 2.87 | 4.00 | 3.576 | 5.065 |
| 10% | 2.53 | 3.59 | 2.977 | 4.260 |

4.3.1. Long-Run Relationships

After the long-run relationships among variables were confirmed, the model results could be inferred. Table 8 shows the coefficients for the variables in the model, along with their level of significance. The model exhibited a positive trend at the 1% level of significance. Furthermore, income was found to have a positive impact on the real exchange rate in the long run, and it was significant at 1%. This positive impact means that increases in income contribute to the appreciation of the real exchange rate. The appreciation of the real exchange rate results in a loss in Saudi Arabia's trade competitiveness, as local commodities become more expensive relative to imported commodities. These results are consistent with those of [2,3,5], who found that income positively impacted real exchange rates. We also found that the impact of price on the real exchange rate was negative in the long run. This negative impact indicates that price increases will make foreign goods less expensive than locally-made goods. This will result in a demand for foreign currency,

and it will depreciate the local currency. These results are consistent with those of [5], who found that increases in the inflation expectation depreciated the real exchange rate.

**Table 8.** Long-run ARDL coefficients.

| Variable | Coefficient | *t*-Statistic | Probability |
|----------|-------------|---------------|-------------|
| C | 8.529832 *** | 7.663873 | 0.0000 |
| @TREND | −0.044433 *** | −6.391937 | 0.0000 |
| LREER(−1) | −0.590601 *** | −6.487778 | 0.0000 |
| LY | 0.543635 *** | 4.500687 | 0.0007 |
| LM2 | 0.034347 | 0.336770 | 0.7421 |
| LP(−1) | −0.168128 ** | −2.319727 | 0.0388 |
| LSDI(−1) | −0.183074 *** | −3.962938 | 0.0019 |
| LOIR(−1) | −0.490167 *** | −7.686131 | 0.0000 |
| LEX(−1) | 0.074998 | 1.052284 | 0.3134 |

** and *** denote statistical significance at the 5%, and 1% levels, respectively.

The impact of sustainable economic development on the real exchange rate was found to be negative in the long run at the 1% significance level. These results indicate that the Saudi Arabian real exchange rate will depreciate as sustainable economic development improves, and these results are consistent with those of [18], who found that Saudi Arabia and the USA had close SDI levels. The increased application of sustainability practices by companies in Saudi Arabia will contribute to increased production costs for different economic sectors (e.g., manufacturing, agriculture, and services), and cost increases in the economic sector will decrease the profitability of companies, which may increase unemployment. Furthermore, the increase in the world's interest in sustainable economic development will reduce the demand for goods produced by unsustainable resources. Therefore, Saudi manufacturers who operate with unsustainable resources will face decreased demand for their produced goods. The sum of these challenges will weaken the exchange rate due to the lowered demand for Saudi Arabian exports and the increased local spending on imported goods and services. On the other hand, the depreciation of the real exchange rate could be a competitive advantage in international trade, although this is not true in the case of sustainable economic development because relative prices are not the main determinant of demand. Furthermore, oil rent was found to have a negative impact on the Saudi real exchange rate in the long run. This negative impact can also be explained through cost increases. Saudi Arabia is mainly an oil exporter, and it imports most of its intermediate and final goods from abroad. Increases in oil prices will have positive impacts on manufacturing and services that use oil. Therefore, import dues in Saudi Arabia will increase, thus leading Saudi Arabia's foreign currency reserves to decrease and have downward pressure on the real exchange rate. Money supply and government expenditure were found to have positive but insignificant trends in the long run.

### 4.3.2. Normalizing Coefficients

The results herein have only been discussed in terms of the direction of the effect, not the size of the impact. Normalization was used to estimate the coefficients by dividing the long-run coefficient by the lag of the dependent variable; accordingly, the dependent variable lag must be significant and negative to achieve normalization. Table 9 shows the normalized long-run coefficients for the independent variables in the model. Our impact analysis of the size coefficients relied on elasticity based on the log form variables used in the estimation. A 1% increase in income was found to contribute a 0.92% appreciation in the real exchange rate at the 1% significance level. The results showed an almost unit elasticity between income and the real exchange rate in Saudi Arabia. However, a price increase of 1% was shown to cause the real exchange rate to depreciate by 0.28% at the 5% significance level. Furthermore, the real exchange rate was shown to depreciate by 0.31% if sustainable economic development increased by 1% at the 1% significance level, and the real exchange rate was found to depreciate by 0.82% at the 1% significance level when the

oil rent increased by 1%. However, income and oil were found to have the most significant long-run impacts on the real exchange rate as a size matter. Finally, money supply and government were shown to be insignificant in the long run.

**Table 9.** Normalizing coefficients.

| Variable | Coefficient | *t*-Statistic | Probability |
| --- | --- | --- | --- |
| LY | 0.920477 | 3.762101 *** | 0.0027 |
| LM2 | 0.058157 | 0.336881 | 0.7420 |
| LP | −0.284673 | −2.567778 ** | 0.0246 |
| LSDI | −0.309979 | −3.428675 *** | 0.0050 |
| LOIR | −0.829946 | −6.315545 *** | 0.0000 |
| LEX | 0.126985 | 1.130986 | 0.2802 |

**, and *** denote statistical significance at the 5%, and 1% levels, respectively.

### 4.3.3. Short-Run Dynamics

The short-run dynamic causality for the OLS equation was examined within the (1, 0, 0, 2, 1, 2, 2) ARDL cointegration framework. Table 10 shows the coefficients for the short-run dynamics of the ARDL cointegration model. The short-run dynamics were confirmed since the error correction term (ECT) was significant and negative. To ensure the convergence of the model toward equilibrium in the long run, the ECT should be negative. Here, the ECT was significant at the 1% level, and the bound *t*-test results showed that the ECT was greater than I(1) at the 1% level. The t-statistic for the ECT was -11.1249, and [33] critical t-value was (−3.96, −5.31) for k = 6. However, the ECT coefficient was −0.5906, which means that any distortion occurring in the short run was adjusted by the model within almost two years. This also ensured that the model would remain stable if shocks disturbed the equilibrium. The price variable was positive in the short run, unlike that in the long run, at the 1% significance level in the first moment and at the 5% significance level in the second moment. This means that price increases in the short run may result in increased manufacturing and exporting, which would increase the demand for local goods. However, in the long run, wage rigidity would be dissolved by new contracts, and prices would again have negative impacts as local prices increase (note that we used price differentials in our calculations). In the long run, sustainable economic development was shown to have a negative impact in the short run, though the short-run impact was relatively smaller than the long-run impact. This means that short-run costs are rigid, and the economy can sustain cost increases in the short run as the sustainability performance increases. The negative impact of sustainable economic development was found to be significant at the 5% level. On the other hand, oil was found to negatively impact the real exchange rate in the first moment of the short run at the 1% significance level. However, in the second moment, the impact of oil was positive and significant at the 1% level. This means oil has an ambiguous impact on the real exchange rate when two forces affect the oil market simultaneously. The first force may be the cost effect, as an increase in oil prices increases all manufacturing costs for goods that use oil as an input. The second force may be the income effect, as demand increases result in oil revenue increases that contribute to the appreciation of the real exchange rate. These results are consistent with those of [12] regarding the first moment in the short run. Unlike in the long run, government expenditure was found to have a positive impact on the real exchange rate in the short run, i.e., an increase in government expenditure will contribute to the appreciation of the real exchange rate. The first and second moments of government expenditure in Saudi Arabia were found to be significant at the 1% level.

**Table 10.** Short-run dynamic coefficients for ARDL approach.

| Variable | Coefficient | *t*-Statistic | Probability |
|---|---|---|---|
| C | 8.529832 *** | 11.06312 | 0.0000 |
| @TREND | −0.044433 *** | −10.55980 | 0.0000 |
| D(LP) | 0.462469 *** | 3.489313 | 0.0045 |
| D(LP(−1)) | 0.439512 ** | 2.467592 | 0.0296 |
| D(LSDI) | −0.048815 ** | −2.297580 | 0.0404 |
| D(LOIR) | −0.309840 *** | −13.40633 | 0.0000 |
| D(LOIR(−1)) | 0.148308 *** | 8.864168 | 0.0000 |
| D(LEX) | 0.205782 *** | 6.291225 | 0.0000 |
| D(LEX(−1)) | 0.156707 *** | 4.869260 | 0.0004 |
| CointEq(−1) | −0.590601 *** | −11.12494 | 0.0000 |

** and *** denote statistical significance at the 5% and 1% level respectively.

The long- and short-run dynamics showed that sustainable economic development negatively impacts the real exchange rate. These results have interesting implications that should be considered when authorities implement a sustainable economic framework, and this study can provide insights regarding the time horizon and challenges that authorities may face when imposing sustainable policies within economic sectors.

## 5. Discussion

The impact of sustainability is negative in the long- and short-run dynamics, emphasizing that Saudi Arabia's dependence on oil is not easy to omit in the near future without scarification. This result is consistent with [18], which is applied to an industrial economy such as the USA. Saudi Arabia's economy relies on oil in different aspects, where oil revenue is the main driver of the economy. Moreover, the economy in Saudi Arabia enjoys low cost of energy as the industrial and household sectors are heavily dependent on carbon-based fuels as a source of energy. This dependence is a result of the advantage Saudi Arabia's economy was gifted by Mother Nature. It is not surprising that Saudi Arabia performs higher on average than the USA in SDI, with a range of less than 50%. However, most of the Gulf Corporation Consul (GCC) countries are lower than Saudi Arabia in their performance in SDI. It may be more interesting to study the impact of SDI on their real exchange rate behavior. Furthermore, selected industrial countries such as Japan, Korea and the UK are performing within the range of Saudi Arabia's performance in SDI. It is worth looking at their real exchange rate behaviors to confirm or contradict the phenomena captured in this study. This outcome has confirmed the association between Sustainability and the real exchange rate in the long- and short-run dynamics where the null hypothesis of no cointegration between the two variables is rejected. On the other hand, income is positively impacting the exchange rate, which consistent with [2,3,5] findings. The main idea of the effect of income on the real exchange rate comes from the relative prices approach. The increase in income will contribute more upward pressure on aggregate demand, and that pressure will finally end in the increase of price level in the economy. Increasing the local price level will change the ratio of the two countries' relative prices in favor of high-income countries. However, the case of Saudi Arabia can be explained via its oil revenue. Thus, Saudi Arabia accumulates foreign currency through oil sales to the rest of the world. The economy achieves its strength by accumulating more foreign currency to satisfy the import of foreign goods and finance future projects, as well as increasing international public fund investment. Furthermore, the result boosts the importance of the income toward the real exchange rate since the coefficient of the income is the highest coefficient in the model. On the other hand, the result of income impacting the real exchange rate is unlike [1]'s finding, and this may be due to the setup of the analysis framework as well as the period in focus. However, the price level needs to have a negative impact in the long run and a positive impact in short-run dynamics to be consistent with [5]'s findings. Furthermore, the impact of the price level in the short-run dynamics is more notable as the highest coefficient for the first moment and the second moment after the oil coefficient. This can support the

substitution effect among the variables where income carries the impact in the long run with a positive direction while prices adjust the exchange rate with a mild effect.

The impact of oil on the exchange rate is negative in both long-run and short-run dynamics, and this can be explained by the result of [11]. The increase in oil will increase the cost of production in developed countries where oil is an important input factor to their production process. Furthermore, an increase in the cost leads to an increase in the prices, which will lower the real exchange rate based on the relative prices approach. However, the result is consistent with [12]'s findings, where oil has a negative impact on the real exchange rate of Saudi Arabia in the long run and short run dynamics. On the other hand, government expenditure has a positive impact in the short-run dynamics, unlike the long-run, where the government expenditure is insignificant. This result is interesting as Saudi Arabia's economy is mainly controlled by the public sector. However, this result could open a wide door for analysis of the private expenditure in the model by decomposing the GDP in its elements. Including the private expenditure could capture the long-run effect on the real exchange rate attributed to different sources of expenditures in the economy. Moreover, this result is an insight that the Saudi Arabian government gives the opportunity to other sectors to lead in the long run to have more sustainability as those sectors operate with market efficiency. On the other hand, money supply is insignificant in affecting the real exchange rate in Saudi Arabia in the long run, unlike [7–9]'s findings, to have a significant impact of money supply on the real exchange rate. However, [7] has found that the money supply is insignificant in affecting the real exchange rate in the short-run dynamics. The result is not a shocking outcome in Saudi Arabia's case, where the monetary authority adopted a bagged exchange rate to the USD as the monetary system of the Saudi riyal.

## 6. Conclusions

The sustainability subject is widely discussed in Saudi Arabia in Vision 2030. Vision 2030 has launched 17 sustainable programs to cover a wide range of Saudi Arabia's economy. These programs will have an impact on different economic factors, such as the real exchange rate. The impact of sustainability on the real exchange rate should be examined to sense the future challenges that could occur to the real exchange rate within the implementation period of those programs. This study assessed the cointegration between the real exchange rate and sustainable economic development in Saudi Arabia in terms of the long-run and short-run dynamics. The study has assumed there is a long-run relationship between sustainable economic development and the real exchange rate in Saudi Arabia. An ARDL cointegration framework was implemented to examine the validity of the study assumption. Our econometric model was constructed with controlled variables based on economics theories in the monetary aspect, as well as our key variable of sustainable economic development. Additionally, macroeconomic factors were implemented in the model based on macroeconomic theories and the literature. These factors were the GDP, money supply in its broad form, the CPI, the Sustainable Development Index (SDI), oil rent, and government expenditure. This study covered the period between 1990 and 2019.

ARDL cointegration confirmed the long-run relationships among the variables. Thus, the assumption of no cointegration has been rejected within 1% significance. Furthermore, the ECT was tested, and it was significant, negative, and did not exceed -1, thus confirming the convergence of the model from the short- to long-run equilibrium. The econometrics outcomes have confirmed the correction of itself within the range of almost two years in case any distortion occurs in the short-run dynamics. The model exhibited a positive trend and was significant at the 1% level. The lag of the real exchange rate was negative and significant at the 1% level. However, income was found to positively impact the real exchange rate in the long run; this result is consistent with the work of [2,3,5]. On the other hand, income was shown to have no impact on the real exchange rate in the short run. The CPI was found to negatively impact the real exchange rate in the long run and positively impact the real exchange rate in the short run. These results are consistent with those of [5], who found that inflation expectations impact the real exchange rate.

Furthermore, we found that sustainable economic development had a negative impact on the real exchange rate in the short and long run. These results are consistent with those of [18], who found that sustainable economic development in the USA has a negative impact on the dollar exchange rate against major world currencies. This result has ensured the relationship between sustainable economic development and real exchange rate where the no cointegration null hypothesis is rejected. Furthermore, the negative sign emphasizes that an increase in the thrust toward more sustainability programs could put more undesirable burden on the real exchange rate. However, oil rent was found to have a negative impact on the real exchange rate in the long run (unlikely in the short run), and the direction of impact was negative in the first moment and positive in the second moment. Furthermore, our findings regarding the impact of oil rent on the real exchange rate were consistent with those of [10], who confirmed the relationship between the two variables. Additionally, the impact of government expenditure was found to be insignificant in the long run and significant in the short run. The impact of government expenditure was positive in the short run, which was consistent with the work of [17,39].

The results of this study are important to the Saudi Arabian authorities who are implementing Saudi Vision 2030. Sustainable economic development is the core of Vision 2030, whose sustainable programs cover a wide range of Saudi Arabian economic sectors. However, the complexity of the real exchange rate and its determinants make it difficult to apply this study's findings to the implementation challenges facing sustainable programs. Furthermore, Saudi authorities should consider the negative impact of the sustainable programs on the real exchange rate, which is relatively stronger in the long run than in the short run. Thus, the transition of the Saudi Arabian economy toward sustainable economic development should use more precautionary policies than relaxed sustainable economic policies. Implementing a new sustainable program should be assessed with different dimension analysis. The inquiries of using renewable sources for energy in factories or investing in a green economy are part of the Saudi Vision 2030 that is in progress and should be applied to selective sectors instead of generalizing the policy over the entire economic sectors. The selection should rely on the size of the impact on the economy with an assessment of risk failure. The sustainability challenges for the Saudi economy are significant, especially because the Saudi economy currently depends on nonrenewable resources. Therefore, it is worth examining the impact of the SDI in different Saudi economic sectors to ensure a broad understanding of the impacts of sustainable economic development on the entire Saudi economy.

**Funding:** The author acknowledges the Deanship of Scientific Research at King Faisal University for the financial support under the Ambitious Researcher Track (Grant No. 3844).

**Institutional Review Board Statement:** Not applicable.

**Data Availability Statement:** The data were acquired from the IMF database accessed in 1/05/2023 under the following link https://data.imf.org/?sk=4c514d48-b6ba-49ed-8ab9-52b0c1a0179b&sId=1390030341854, World Bank database accessed on 1 May 2023 under the following link https://databank.worldbank.org/reports.aspx?source=2&country=SAU and Sustainable Development Index accessed on 1 May 2023 under the following line https://www.sustainabledevelopmentindex.org/time-series.

**Conflicts of Interest:** The author declares no conflict of interest.

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
