# Peer review of "The Impact of a Sustainable Economic Development Focus on the Real Exchange Rate in Saudi Arabia"

_sustainability, doi:10.3390/su151813422_

Round 1

Reviewer 1 Report

I appreciate the documentation made to highlight the current state of knowledge in the targeted research field.

I also appreciate the application part through which the quantitative econometric models are validated, including the conclusions at the end.

Consequently, I appreciate the effort put into writing this article proposal and I recommend its publication.

Author Response

Dear Reviewer

Hope you are doing fine. I would like to thank for your efforts and comments on the article. Also, I appreciate your interest toward the subject of the article.

Reviewer 2 Report

Dear author,

Thank you for sending your paper to the journal. The paper needs a major revision in the following areas:

1-The abstract should cover: purpose, method, findings and originality

2-Introduction is very short and this part should address the research problem and research gap and the originality of the paper

3-There are no theoretical issues on the paper and this part should be included aims to support the models

4- It is recommended to include more control variables to get better results

5-In the conclusion part you repeated the results and you need to bring your results with different words and you need to conclude according to the findings

6- Do not forget the discussion section

7-What are the practical and managerial implications?

Needs minor revision

Author Response

Hope you are doing fine. I would like to thank for your efforts and comments on the article. I have gone over the comments and corrected the article accordingly.  

1-The abstract should cover: purpose, method, findings and originality

I have revised the abstract to reflect your concern regarding the missing elements. I have included purpose, method, findings and originality with highlighted all changes in yellow.

2-Introduction is very short and this part should address the research problem and research gap and the originality of the paper

I have included research problem and represent the research gap as well the originality of the article. All changes are highlighted in yellow.   

3-There are no theoretical issues on the paper and this part should be included aims to support the models

I have added conjunction statement between the method and the aim of the article. All changes are highlighted in yellow.

4- It is recommended to include more control variables to get better results

The control variables have been included based on monetary and macroeconomics theories. Adding more variables would have singularity in the estimation specially with short period data which the article dealing with. I have stated that as extension, it would be useful to have decomposed data for the GDP to distinguish between private and public expenditure.      

5-In the conclusion part you repeated the results and you need to bring your results with different words and you need to conclude according to the findings

Change have been made in the conclusion and highlighted with yellow. Hope these changes make the conclusion more motivated.

6- Do not forget the discussion section

The discussion section has been added and thanks for this comment. All discussion section is highlighted with yellow.

7-What are the practical and managerial implications?

I have stated some of the practical and suggestion regarding new sustainable programs. These suggestions are in general as well strategic plan of the implementation.   

I would like to thank you for your efforts and your comments are appreciated.

Reviewer 3 Report

The subject of the paper treats the Impact of a Sustainable Economic Development Focus on the Real Exchange Rate in Saudi Arabia.  This paper will be considered for publication but after a major revision taking account of the following comments and suggestions.

The introduction is very short. The authors must rewrite it as this introduction is weak and does not highlight the importance of the research in terms of the contribution and the general framework of the research and the research question...

The choice of Saudi Arabia as a case study should be motivated.

Authors must detail the gap in their research and highlight the novelty of their study and should do the mention in the Abstract and Introduction sections.

In introduction, the authors should explain why they used ARDL. To be specific, the advantages of ARDL should be mentioned. They should provide some literature which has used a similar method before.

The analysis of the results and the manner of delivering them does not provide accurate evidence through the manuscript. Please analyze and detail the mechanism underlying the results and compare them among similar research in the field.

The authors should discuss the empirical results with the existing field literature.

Moderate editing of English language required

Author Response

Hope you are doing fine. I would like to thank for your efforts and comments on the article. I have gone over the comments and corrected the article accordingly.

The introduction is very short. The authors must rewrite it as this introduction is weak and does not highlight the importance of the research in terms of the contribution and the general framework of the research and the research question...

I have revised the introduction to reflect the contribution of the article and represent the importance aspects of the study. Also, research question as well methodology have been addressed. All changes are highlighted in yellow.

The choice of Saudi Arabia as a case study should be motivated.

I have stated why Saudi Arabia has chosen in the introduction with highlighting the changes in yellow.  

Authors must detail the gap in their research and highlight the novelty of their study and should do the mention in the Abstract and Introduction sections.

I have rewritten the abstract and the introduction to reflect your comments with yellow highlighting.  

In introduction, the authors should explain why they used ARDL. To be specific, the advantages of ARDL should be mentioned. They should provide some literature which has used a similar method before.

I have stated that there is two method and the selection of those one of those two methods depends on the property of the data. Also, I mentioned that in the literature is more common to use integration method for such analysis. However, adding more article in the introduction and repeat them in the literature would have over sighting to the readers. With all respect to your comment in this aspect, my point of view is making the selection of the method to be based on scientific process rather than literature based. Furthermore, in the methodology section I have showed why I selected the ARDL cointegration method.        

The analysis of the results and the manner of delivering them does not provide accurate evidence through the manuscript. Please analyze and detail the mechanism underlying the results and compare them among similar research in the field.

I have added a discussion section to reflect your comment in subject and all discussion section is highlighted in yellow.   

The authors should discuss the empirical results with the existing field literature.

I have discussed the result with literature and have comparison between the outcome of the article and the findings in the literature. All those discussions have been add in the discussion section with yellow highlighting.

I would like to thank you for your efforts and your comments are appreciated.

Reviewer 4 Report

This is another replication study with some noteworthy innovations by the author. The author assessed the co-integration between the real exchange rate and sustainable economic development in terms of dynamics. The econometric model was constructed by the author with controlled variables, as well as with the key variable of sustainable economic development. The macroeconomic factors used by the author in the model are the money supply in its broad form, the Sustainable Development Index, oil revenues and government spending. The author tested the ECT and found it to be significant, negative. The author stated that income has a positive effect on the real exchange rate in the long run and that income has no effect on the real exchange rate in the short run. The author concluded that the CPI had a negative impact on the real exchange rate in the long term and a positive impact on the real exchange rate in the short term. The author stated that sustainable development negatively affects the real exchange rate in the short and long term. The literature should be supplemented with research on countries where the policy of sustainable development is systematic and long-term, and explain why sustainable development affects the real exchange rate in the short and long term in a different way in this type of country than the author did.

Author Response

Hope you are doing fine. I would like to thank for your efforts and comments on the article.

This is another replication study with some noteworthy innovations by the author. The author assessed the co-integration between the real exchange rate and sustainable economic development in terms of dynamics. The econometric model was constructed by the author with controlled variables, as well as with the key variable of sustainable economic development. The macroeconomic factors used by the author in the model are the money supply in its broad form, the Sustainable Development Index, oil revenues and government spending. The author tested the ECT and found it to be significant, negative. The author stated that income has a positive effect on the real exchange rate in the long run and that income has no effect on the real exchange rate in the short run. The author concluded that the CPI had a negative impact on the real exchange rate in the long term and a positive impact on the real exchange rate in the short term. The author stated that sustainable development negatively affects the real exchange rate in the short and long term. The literature should be supplemented with research on countries where the policy of sustainable development is systematic and long-term, and explain why sustainable development affects the real exchange rate in the short and long term in a different way in this type of country than the author did.

The method has been used in this study is wildly used in the literature for assessing the relationship among the variables as well the impact of independent variables on dependent variable. However, the other study that been done was discussing different problem with some of the variables not all. Furthermore, this study tries to touch the base of the sustainability in a country that known to be one of the main sources of energy based on oil. Saudi Arabia has launched its vision for the coming decade to be known as 2030 vision with high impassion toward changing the structural of the economy from oil based to renewable energy-based economy. This political decision worth to look at the challenges that Saudi economy may face in the future. However, there is no innovation in the econometrics side which is applicable to the most of the applied economics researches, but the outcomes have an opportunity to be discussed specially in term of the risk that associated with such critical decision. I hope with the changes that I have made to the study as a response to the reviewers’ comments will bring a motivation to the article make it sound. Also, with more attention to sustainability that will motivate econometrician to invent new econometrics method to enrich the field. In term of your comment regarding the literature of the sustainability, there is lack of studies in this subject and this is one of the motivations for this study. The SDI has been published in 2020 and the subject it’s new in economics field thus little to be found discussing the impact of the sustainability.         

I would like to thank you for your efforts and your comments are appreciated.

Reviewer 5 Report

Dear Author

Your article is very interesting, and I am grateful for the opportunity to read it. I think that subject of the research is interesting, and the results of the research give a lot of new information and possibilities of further analysis.

Reading the text, I found a few elements that I think would improve your article.

1.: There is no clear presented assumptions, hypothesis, or research questions. The aim is not enough to establish a consistent and logical path between the main goal and the research results.

You should show your logical path of thinking in the first part of your article. The methodology is the description of the research path only but whole article it is something more than research. I think is necessary step to complete this part of your article.

2.: The results are too elaborate, if you present everything, that is, every type of exploration, you generate a chaotic effect in the article. You should select the most important tables and limit the presentation to them. What is left should only be described. For readers, every step you have taken is not important. Readers are looking for results, examples, opinions and methods, and you should show this. If you leave it in your article, there will be a huge imbalance between chapters.

3.: There is no discussion. I think about the discussion as an attempt to confront your opinion with another and The Discussion as a chapter. So, I miss both.

To increase the significance of the results, the discussion should include the differences and similarities among your findings and those of other scholars. The analysis of other studies and analyzes is clearly missing. Just as there is no reference to other points of view, different situation, and examples. There is no application of a scientific analysis to reality and no real discussion. All this consequently reduces the article to the research part and causes the reader to judge the quality of the research but not its purpose or conclusions.

4.: The Conclusions chapter is the weak part of your article. In my opinion conclusions are insufficient - The conclusion section should be a brief summary of article’s aim, methods and findings. But it's not here. This chapter should be extended. For me, the summary is too limited, there is no reference to your assumptions, your hypothesis or research questions. At this point, you should show references to your research and all formal aspects of your article. At the begging and at the end you should include a description of the research questions and hypotheses. Develop and explain goals. It is necessary to change the convention from the presentation of research to the presentation of results and conclusions. In general, I believe that hypotheses and research questions should be presented. The goals should be presented and explained. At the end, the conclusions should refer to each of the goals.

5.: I saw it in the comment that you use sustainable development and sustainable economic development interchangeably – it could be confusing.

Summarizing.

I find your article very good. I really like your article and appreciate your work. It is interesting topic, and the conclusions could open the way for further research. You have to make some changes especially in Results, Discussion and Conclusions. But my general opinion and my assessment of your research and whole article is positive.

Good luck! 

Author Response

Hope you are doing fine. I would like to thank for your efforts and comments on the article. I have gone over the comments and corrected the article accordingly.

1.: There is no clear presented assumptions, hypothesis, or research questions. The aim is not enough to establish a consistent and logical path between the main goal and the research results.

You should show your logical path of thinking in the first part of your article. The methodology is the description of the research path only but whole article it is something more than research. I think is necessary step to complete this part of your article.

I have established a conjunction between the aim of the article and the result by connecting the outcomes with hypothesis. All changes in this aspect have been highlighted in yellow.

2.: The results are too elaborate, if you present everything, that is, every type of exploration, you generate a chaotic effect in the article. You should select the most important tables and limit the presentation to them. What is left should only be described. For readers, every step you have taken is not important. Readers are looking for results, examples, opinions and methods, and you should show this. If you leave it in your article, there will be a huge imbalance between chapters.

I have made some focus on sustainability to be represented with more details.  Also, all changes in this aspect are highlighted in yellow.

3.: There is no discussion. I think about the discussion as an attempt to confront your opinion with another and The Discussion as a chapter. So, I miss both.

To increase the significance of the results, the discussion should include the differences and similarities among your findings and those of other scholars. The analysis of other studies and analyzes is clearly missing. Just as there is no reference to other points of view, different situation, and examples. There is no application of a scientific analysis to reality and no real discussion. All this consequently reduces the article to the research part and causes the reader to judge the quality of the research but not its purpose or conclusions.

I have added a discussion section and is highlighted with yellow.  

4.: The Conclusions chapter is the weak part of your article. In my opinion conclusions are insufficient - The conclusion section should be a brief summary of article’s aim, methods and findings. But it's not here. This chapter should be extended. For me, the summary is too limited, there is no reference to your assumptions, your hypothesis or research questions. At this point, you should show references to your research and all formal aspects of your article. At the begging and at the end you should include a description of the research questions and hypotheses. Develop and explain goals. It is necessary to change the convention from the presentation of research to the presentation of results and conclusions. In general, I believe that hypotheses and research questions should be presented. The goals should be presented and explained. At the end, the conclusions should refer to each of the goals.

I have made changes in the summery to reflect the aim of the article, methodology and findings. Also, I connected the finding with the hypothesis of the study. All changes are highlighted in yellow.   

5.: I saw it in the comment that you use sustainable development and sustainable economic development interchangeably – it could be confusing.

I have corrected the article to reflect sustainability economic development to be consistent allover the article. However, the only use of sustainable development when it is mentioned the index.   

I would like to thank you for your efforts and your comments are appreciated.

Round 2

Reviewer 2 Report

Dear author,

Thank you for sending your revised paper; you amended my comments in the current version.

Author Response

Dear Reviewer

Hope you are doing fine. I would like to thank for your efforts and comments on the article. Also, I appreciate your time that you contributed to help in improving the article.

Reviewer 3 Report

The introduction is still inadequate. The effort made in emphasizing the benefits of choosing the Kingdom of Saudi Arabia falls short of expectations. A thorough examination of the grounds for selecting Saudi Arabia is required.

I also request that the author emphasize the desired contribution of this research in comparison to past research.

Moderate editing of English language required

Author Response

Hope you are doing fine. Kindly, the response for your comments is attached.

The introduction is still inadequate. The effort made in emphasizing the benefits of choosing the Kingdom of Saudi Arabia falls short of expectations. A thorough examination of the grounds for selecting Saudi Arabia is required.

I have mentioned the importance of Saudi Arabia within the context of the world as well its core role in oil market as major producer. However, I touched the based that Saudi Arabia is a member in G20 as well a potential member in BRICS. Those two global organizations have precise interest toward sustainability and implementing standards with insurance of compliance. All those standards are adapting HDI standards. However, Saudi Arabia is ranked 18th GDP in the world. All those information characterized Saudi Arabia interesting country for sustainable studies.  All changes highlighted in red.                        

I also request that the author emphasize the desired contribution of this research in comparison to past research.

I have written a statement in the end of the literature review showing what is the desire of this study that focusing in developing country with dependency on natural resources as main income for the economy. All changes are highlighted in red.

The thanks for your valuable comments that improving this study. Also, I would like to thank you for your time that you contributed in this study with your aim to be improved. Your efforts are appreciated.

Reviewer 4 Report

-

Author Response

Hope you are doing fine. I would like to thank you for your efforts and your comments are appreciated.

Round 3

Reviewer 3 Report

Accept after minor revision

Minor editing of English language required